# Optimization of In Vitro Culture Conditions of Sturgeon Germ Cells for Purpose of Surrogate Production

**DOI:** 10.3390/ani9030106

**Published:** 2019-03-21

**Authors:** Xuan Xie, Ping Li, Martin Pšenička, Huan Ye, Christoph Steinbach, Chuangju Li, Qiwei Wei

**Affiliations:** 1Key Laboratory of Freshwater Biodiversity Conservation, Ministry of Agriculture of China, Yangtze River Fisheries Research Institute, Chinese Academy of Fishery Sciences, Wuhan 430223, China; xxie@jcu.cz (X.X.); yehuan85@163.com (H.Y.); lcj@yfi.ac.cn (C.L.); 2Research Institute of Fish Culture and Hydrobiology, University of South Bohemia in České Budějovice, Faculty of Fisheries and Protection of Waters, South Bohemian Research Center of Aquaculture and Biodiversity of Hydrocenoses, Zátiší 728/II, 38925 Vodňany, Czech Republic; liping2018@email.sdu.edu.cn (P.L.); psenicka@frov.jcu.cz (M.P.); steinbach@frov.jcu.cz (C.S.); 3Marine College, Shandong Universit, Weihai 264209, China; 4Sino-Czech Joint Laboratory for Fish Conservation and Biotechnology, Yangtze River Fisheries Institute, Chinese Academy of Fishery Sciences, Wuhan 430223, China

**Keywords:** feeder cells, germ cell culture, glial-cell-derived neurotrophic factor, sturgeon, transplantation

## Abstract

**Simple Summary:**

The sturgeon is among the most ancient of actinopterygian fishes. Most species of sturgeon are listed as critically endangered due to habitat alteration caused by damming of rivers, pollution and overharvesting. Germ cell transplant is a useful tool to save these endangered species. To expand germ cell populations and sustain the supply for long periods for transplant, we established basal culture conditions for sturgeon germ cells. Germ cell mitotic activity has been enhanced by eliminating gonad somatic cells, supplementing with growth factor and using an alternative to fetal bovine serum. The optimal condition identified was purified germ cells cultured in serum-free medium supplemented with leukemia inhibitory factor (LIF) and glial cell line-derived neurotrophic factor (GDNF) at 21 °C. Cultured sterlet germ cells showed development after transplant into Russian sturgeon. The study provided useful information for sturgeon germ cell culture.

**Abstract:**

To expand germ cell populations and provide a consistent supply for transplantation, we established basal culture conditions for sturgeon germ cells and subsequently increased their mitotic activity by eliminating gonad somatic cells, supplementing with growth factor, and replacing fetal bovine serum (FBS). The initial basal culture conditions were Leibovitz’s L-15 medium (pH 8.0) supplemented with 5% FBS (*p* < 0.001) at 21 °C. Proliferation of germ cells was significantly enhanced and maintained for longer periods by elimination of gonad somatic cells and culture under feeder-cell free conditions, with addition of leukemia inhibitory factor and glial-cell-derived neurotrophic factor (*p* < 0.001). A serum-free culture medium improved germ cell proliferation compared to the L-15 with FBS (*p* < 0.05). Morphology remained similar to that of fresh germ cells for at least 40 d culture. Germline-specific gene expression analysis revealed no significant changes to germ cells before and after culture. Sterlet *Acipenser ruthenus* germ cells cultured more than 40 days showed development after transplant into Russian sturgeon *Acipenser gueldenstaedtii*. Polymerase chain reaction showed 33.3% of recipient gonads to contain sterlet cells after four months. This study developed optimal culture condition for sturgeon germ cells. Germ cells after 40 d culture developed in recipient gonads. This study provided useful information for culture of sturgeon germ cells.

## 1. Introduction

Germ stem cells have the ability to self-renew as well as to differentiate into other germ cell stages. Xenotransplantation of germ cells has been conducted in fish [1]. In contrast to mammals, in fish, both type A spermatogonia and oogonia (probably germ stem cells) show high sex plasticity even after sexual maturation [2]. Germ cells cultured in vitro could be transplanted in cases of limited numbers of an individual’s own germ cells, low stem cell purification efficacy, and cells damaged by enzymatic dissociation [3]. In recent years, in vitro culture of germ cells was established in Medaka *Oryzias latipes* [4], zebrafish *Danio rerio* [5], Nile tilapia *Oreochromis niloticus* [6] and rainbow trout *Oncorhynchus mykiss* [7].

Sturgeons belong to the order Acipenseriformes, which are among the most ancient of actinopterygian fishes [8]. According to the International Union for Conservation of Nature and Natural Resources’ Red List, 64% of sturgeon species are critically endangered due to habitat alteration caused by damming of rivers, pollutio, and overharvesting [9,10,11]. Most sturgeon species are late maturing, making culture and conservation costly and time consuming [12,13]. Germ cell culture and transplant could be an available and rapid method for surrogate production of endangered fishes with large bodies and a long life-cycle. To establish optimal culture conditions for sturgeon germ cells and improve their mitotic activity, we investigated the basal culture conditions for gonad cells and examined the effect of somatic cells on germ cell proliferation and assessed the influence of growth factor on germ cell mitotic activity. The L-15 modified culture medium with fetal bovine serum (FBS) was replaced with a serum-free medium. The identity of cultured germ cells was confirmed by RT-qPCR (Quantitative real-time PCR) targeting germ cell specific genes, and the cells were transplanted into sturgeon larvae to assess their transplantability and proliferation.

## 2. Materials and Methods

### 2.1. Animal Ethics Statement

Animal handling and experimentation were approved by the Ethics Committee on Animal Care of Chinese Academy of Fishery Science and the Ministry of Agriculture of the Czech Republic (reference number: 53100/2013-MZE-17214).

### 2.2. Fish Selection and Sampling

Dabry’s sturgeon *Acipenser dabryanus*, used for optimization of germ cell culture conditions, were reared in the Taihu Station, Yangtze River Fisheries Research Institute, Chinese Academy of Fisheries Science. Sterlet *Acipenser ruthenus* used for germ cell transplantation were cultivated at the Faculty of Fisheries and Protection of Waters, University of South Bohemia. Gonads were collected from 22–26-month-old Dabry’s sturgeon (length ~92 cm; weight ~3.5 kg). Sterlet gonads were collected from 10–13-month-old specimens (~52 cm; ~520 g). The gonads were at maturity stage II: containing mostly spermatogonia or oogonia and previtellogenic oocytes. Deep anesthesia was induced by 0.05% 3-aminobenzoic acid ethyl ester methanesulfonate-222 (MS-222) (Sigma, St. Louis, MO, USA). Russian Sturgeon *Acipenser gueldenstaedtii* larvae obtained from combined eggs and sperm of three females and three males were used as recipients for cultured germ cells.

### 2.3. Dissociation and Culture of Gonad Cells

Gonads of Dabry’s sturgeon were washed in phosphate-buffered saline (PBS; Sigma-Aldrich, St Louis, MO, USA) containing 50 μg/mL ampicillin, 200 U/mL penicillin, and 20 μg/mL streptomycin (Sigma) (pH 8.0) and minced into 1-mm^3^ pieces. Fragments were dissociated using various proteinases with gentle pipetting. For all experiments, cells were seeded at a concentration of 1.6 × 10^4^–2 × 10^4^ cells/cm^2^ in 25-cm^2^ culture flasks containing 5 mL culture medium.

### 2.4. Optimization of Basal Culture Conditions

To assess the effect of enzymes on germ cell mitotic activity, gonads were dissociated with one of three enzyme treatments: (1) 0.47% trypsin–Ethylenediaminetetraacetic acid (trypsin–EDTA; Gibco, Grand Island, NY, USA) digestion for 15 min with gentle pipetting; (2) 0.25% trypsin (Worthington Biochemical Corporation, Lakewood, NJ, USA) digestion for 3 h; or (3) a combination of 2 mg/mL collagenase H (Roche Diagnostics, Mannheim, Germany) and 500 U/mL Dispase II (Sigma-Aldrich, St Louis, MO, USA) digestion for 3.5 h, as previously described [14]. The digestion was stopped by Leibovitz’s L-15 medium (Sigma-Aldrich, St Louis, MO, USA) with 20% FBS, filtered through a 40-μm pore-size nylon screen, and centrifuged at 200× *g*. The dissociated cells were cultured in L-15 medium supplemented with 15% (*v/v*) FBS, 25 mM HEPES (Sigma-Aldrich, St Louis, MO, USA), and antibiotics at 25 °C. Cells were cultured for 10 d, with 5-Bromo-2-deoxyUridine (BrdU, Sigma-Aldrich, St Louis, MO, USA) and immunocytochemical assays conducted on the final day. To investigate the influence of temperature, the dissociated cells were cultured in L-15 medium supplemented with 15% FBS, and cultured at 17, 21, and 25 °C. The BrdU and immunocytochemical assays were performed on day 10.

Based on these investigations, the optimal concentration of FBS was determined. The selected culture medium was L-15 supplemented with 5, 10, or 15% (*v/v*) FBS. Cells were cultured at 21 °C. BrdU and immunocytochemical assays were performed on days 5, 10, 15, and 20.

### 2.5. Elimination of Gonad Somatic Cells and Effect of Feeder Layer on Germ Cell Mitotic Activity

The survival and proliferation of germ cells was inhibited by over-growth of gonad somatic cells. We attempted to eliminate somatic cells and assess the influence of the addition of a feeder layer to germ cells. The dissociated gonad cells were cultured for 36 h, and floating cells were gently pipetted and passaged two or three times to obtain purified germ cells.

The feeder cells were fibroblastoid somatic cells derived from sturgeon gonad: Dissociated gonad cells were cultured with L-15 supplemented with 10% (*v/v*) FBS. After three or four passages, the cells were treated with a 10 μg/mL mitomycin C solution for 3 h to arrest growth and then washed three times with L-15. Purified germ cells were transferred onto feeder cells. For feeder-free culture, purified germ cells were transferred directly into a new culture flask. Dissociated non-purified gonad cells were used as controls. The three groups were cultured in L-15 and 5% FBS at 21 °C. Germ cell and somatic cell proliferation was analyzed by BrdU and immunocytochemical assays after 7, 14 and 21 d.

### 2.6. Effects of Growth Factor on Germ Cell Proliferation

Selected growth factors were added at the following concentrations: 25 ng/mL epidermal growth factor (EGF, Pufei, Shanghai, China); 25 ng/mL basic fibroblast growth factor (bFGF, Pufei, Shanghai, China); 5 IU/mL human chorionic gonadotropin (hCG, Sigma-Aldrich, St Louis, MO, USA); 2 IU/mL pregnant mare serum gonadotropin (PMSG, Sigma-Aldrich, St Louis, MO, USA); 25 ng/mL glial cell line-derived neurotrophic factor (GDNF, Thermo Scientific, Waltham, MA, USA); and 25 ng/mL leukemia inhibitory factor (LIF, Peprotech, Rocky Hill, NJ, USA). As Sakai et al. [5] reported that 7% carp serum enhanced the proliferation of germ cells derived from zebrafish, we also evaluated the effect of sturgeon serum. The serum was derived from blood of an 18-month-old male sturgeon, centrifuged at 1000× *g* and filtered through a 0.2-μm pore nylon screen. Serum (7% *v/v*) was added to the L-15 and 5% FBS culture medium. Growth factor was added to the culture medium as follows: (1) EGF and bFGF; (2) hCG and PMSG; (3) EGF, bFGF, hCG and PMSG; (4) Sturgeon serum; (5) GDNF and LIF.

Based on results of the previous step, purified germ cells were seeded in culture flasks and cultured at 21 °C. The mitotic activity of germ cells and somatic cells was investigated by BrdU and immunocytochemical assay on day 10 of culture.

### 2.7. Replacement of FBS

We attempted to establish a serum-free medium to assess the effect of FBS and determine a potential alternative. The serum-free medium comprised StemPro-34 SFM (Invitrogen, Carlsbad, CA, USA) supplemented with StemPro Supplement (Invitrogen, USA). The FBS was replaced with 0.5% (*w/v*) bovine serum albumin (BSA; Sigma-Aldrich, St. Louis, MO, USA), 2 mM L-glutamine (Sigma-Aldrich, St Louis, MO, USA), 0.5% (*v/v*) B27 Supplement (Gibco, USA), 20 μg/mL L-Lys, 20 μg/mL L-Pro, 20 μg/mL L-Asp, and 10 mM/L sodium pyruvate (Sigma-Aldrich, St Louis, MO, USA). We also added 25 mM HEPES, 50 μg/mL ampicillin, 200 U/mL penicillin and 20 μg/mL streptomycin (pH 8.0). Purified germ cells were cultured at 21 °C with serum-free medium and optimal growth factors. The mitotic activity of germ cells and somatic cells was investigated by BrdU and immunocytochemical assay after 10 d culture.

### 2.8. Analysis of Germ Cell Mitotic Activity

To detect germ cell mitotic activity, a BrdU incorporation assay was performed by adding 25 μM BrdU (Sigma-Aldrich, USA) to the culture medium during the final 24 h of culture. The cells were fixed by 4% paraformaldehyde for 30 min. Immunocytochemical detection of BrdU was performed with mouse anti-BrdU antibody (ab8039; Abcam, Cambridge, MA, USA) and Alexa-Fluor-488-labeled Goat Anti-Mouse IgG (H+L) (Beyotime Biotechnology, Hangzhou, China). The germ cells were identified with anti-Vasa antibody from sturgeon [15] (diluted 1:500) and exposed for 1 h to fluorescein-conjugated Alexa-Fluor-647-labeled Goat Anti-Rabbit IgG (H+L) (diluted 1:500). The cells were observed with an inverted fluorescent microscope (Leica, Wetzlar Germany). The percentage of BrdU-labeled germ cells was calculated from randomly selected 500 BrdU-labeled cells.

### 2.9. Expression Analysis

To analyze the fate of the cultured cells, we investigated the presence of germ-line-specific gene transcripts *dead end*, *gfra1a*, *grip2*, *plk3*, and *ednrba*. Total RNA of each sample was reverse transcribed with the TATAA Biocenter Kit. Primers (Table 1) were designed based on the sturgeon gonad transcriptome. Cell expression of selected genes before and after culture was analyzed using quantitative real-time polymerase chain reaction (PCR) performed with SYBR Green Real-time PCR Master Mix according to the manufacturer’s protocol. PCR amplification was conducted with an initial activation of 3 min at 95 °C followed by 40 cycles of 95 °C for 10 s, 60 °C for 20 s, and 70 °C for 30 s. The fluorescent SYBR Green signal was detected immediately after the extension step of each cycle. Gene expression was compared by 2^−ΔΔCq^.

### 2.10. Cultured Germ Cell Transplantation

To examine the proliferation of cultured germ cells in recipient gonads, xenotransplantation was performed. Russian sturgeon larvae two-weeks post-hatching with fluorescein isothiocyanate (FITC)-labeled endogenous primordial germ cells were used as recipients, as described by Pšenička [16]. Culture medium as described (Section 2.7) was used in sterlet germ cell culture for more than 40 d. Paraffin sections of testes and ovaries used are shown in Figure 1A, B. Before transplant, cultured cells were digested by trypsin and resuspended. The germ cell ratio was calculated by immunolabeling. Membranes of both cell types were labeled by the PKH26 Red Fluorescent Cell Linker Kit according to the manufacturer’s protocol (final dye concentration 2 µL/mL). The labeled cells were injected into the body cavity of the host using a micromanipulator M-152 (Narishige, Japan) and a microinjector Cell Tram Vario (Eppendorf, Germany) at ~2.5 × 10^3^ cells per recipient. The larvae were maintained at 18 °C and fed on *Tubifex*. Larvae were examined by fluorescence stereomicroscope at 40 days post-transplant (dpt).

After four months, fish were dissected under stereomicroscope. DNA was extracted from gonads using Exgene^TM^ Genomic DNA micro kit according to manufacturer’s instructions (Gene All Co. Exgene^TM^, South Korea). Sterlet-specific primers (Table 2) were applied for identification of sterlet cells as described by Havelka et al. [17]. The PCR conditions were initial denaturation at 94 °C for 1 min, followed by 35 cycles of denaturation at 94 °C for 15 s, annealing at 50 °C for 15 s, and extension at 72 °C for 30 s.

### 2.11. Statistical Analysis

The results were expressed as a ratio against control as mean ±  Standard Error of Mean (SEM). Significance was determined with one-way and two-way analysis of variance (ANOVA) followed by Turkey tests (see Section 3.1, Section 3.3 and Section 3.4). A paired T-test was used to compare two groups (see Section 3.5). Probability values of *p* < 0.05 were considered significant.

## 3. Results

### 3.1. Optimal Basal Culture Conditions

Results of BrdU assays are shown in Figure 2. Germ cells digested with 0.47% trypsin-EDTA showed significantly lower proliferation rate than the other two groups (*p* < 0.05, Figure 3A). The mitotic activity of gonad somatic cells in the 0.25% trypsin-treated group was significantly higher than that seen in the other two groups (*p* < 0.05, Figure 3B). Although statistical analysis did not reveal significant differences in proliferation rates for germ cells with dissociation method, the mean value for 0.25% trypsin was higher; therefore, we used it in subsequent trials. The mean value of germ cell proliferation was higher at 21 °C than at 17 and 25 °C, but somatic cells cultured at 25 °C showed a significantly higher proliferation rate (*p* < 0.05, Figure 3C,D). The mitotic activity of germ cells with different concentrations of FBS were analyzed as proportion relative to controls, using a two-way ANOVA with concentration and cultured days as the dependent variables. Significant differences in mitotic activity were seen among germ cell groups treated with 5%, 10%, and 15% FBS (*p* < 0.001). The proportion of germ cells with BrdU incorporated was highest when cultured in L-15 medium containing 5% FBS (Figure 3E). Proliferation of germ cells differed significantly at 5 d, 10 d, 15 d, and 20 d (*p* < 0.001) with highest proliferation observed at 10 d culture. The gonad somatic cells showed an opposite trend to that of germ cell proliferation (Figure 3F). Immunolabeling after digestion indicated 44.4% ± 9.81% of cells in testes to be germ cells and 37.5% ± 9.66% of those in ovaries to be germ cells.

### 3.2. Effect of Feeder Layer on Germ Cell Mitotic Activity

Two-way ANOVA, using quantity of germ cells with BrdU incorporated as the dependent variable, and *culture day* and *treatment* as factors, revealed significant variation (*p* < 0.001) in the proliferation of germ cells with both factors (Figure 4). Day and treatment interaction was also significant (*p* < 0.001). Of the groups with and without feeder cells and control, significantly greater proliferation was observed in those cultured without the feeder layer. In control group, germ cell proliferation was decreased after 21 d culture compared to that at 14 d, while somatic cells maintained stable mitotic activity during the same period (Figure 5A,B). In the feeder-free culture, the suspended germ cells tended to form clumps at 3 d culture after purification. The clumps became larger during 3–14 d of culture (Figure 5C,D). Germ cell propagation was reduced after 21 d culture compared to that at 14 d, and the clumps gradually ceased expanding. In the feeder layer group, some cells attached to the feeder cells and were observed to proliferate. There was no further cell attachment to the feeder cells after 2 d. Germ cell growth took place during 3–14 d of culture (Figure 5E,F). Cells were shed from the feeder layer, and the number of germ cells after 21 d culture was lower than that seen at 14 d.

With and without feeder cells, BrdU incorporation showed that mitotic activity of germ cells was highest at 14 d culture and decreased thereafter. The residual somatic cells showed stable growth after 14 d culture. During the same period, the total number of germ cells decreased gradually, although though mitosis was confirmed by BrdU assay.

### 3.3. Effect of Growth Factors on Germ Cell Mitotic Activity

There were significant differences in mitotic activity of germ cells treated with different growth factors (*p* < 0.05, Figure 6A). Addition of LIF and GDNF to the culture medium significantly increased germ cell mitotic activity, with BrdU incorporation in germ cells 1.8-fold that of cells cultured without growth factor after 10 days culture. The cultured germ cells proliferated and formed large clumps over a period of 28 days. Sturgeon serum (7%) was associated with germ cell mitotic activity 6-fold that seen in control conditions. With addition of hCG and PMSG, BrdU incorporation in germ cells was similar to growth-factor-free conditions. Addition of LIF, GDNF, or sturgeon serum was not shown to contribute to somatic cell proliferation (*p* < 0.05, Figure 6B).

### 3.4. Replacement of FBS

BrdU incorporation under serum-free conditions showed significantly higher germ cell mitotic activity than seen with L-15 (*p* < 0.05, Figure 6C,D). After 34 days, the germ cells cultured in serum-free medium were proliferated and tended to form large clumps. Their morphology after at least 38 d culture remained similar to that of fresh germ cells. The number of cells after 40 d culture was 9.6-fold the initial number.

### 3.5. Expression and Transplant Effectiveness of Cultured Germ Cells

We maintained the cells in serum-free medium for 10 days and compared with fresh purified germ cells to investigate sturgeon germ cell fate in our culture system. qPCR analysis confirmed that cultured cells transcribed the germ cell marker genes *dead end*, *grip2*, *plk3, gfra1a*, and *ednrba*. Most marker gene expression showed no significant difference before and after 10 d culture with growth factor (*p* > 0.05, Figure 7), confirming that cultured cells remained similar to fresh germ cells.

To examine whether the transplanted germ cells could develop in recipient gonads, cells cultured for more than 40 d under serum-free conditions were injected into the body cavity of Russian sturgeon larvae. Before transplant, immunolabeling indicated that more than 91% of injected cells were germ cells (Figure 8A–L). Recipient fish were dissected one-month post-transplant. In 50%, attachment and development of the cultured cells in the recipient genital ridge could be detected (Figure 8M,N). After four months, PCR showed 33% of recipient fish to display sterlet-specific bands. In all gonad samples, fragments from both donor and recipient could be detected, which matched the fin tissues of the two species (Figure 8O).

## 4. Discussion

Brinster and Zimmermann [18] first described germ cell transplant in a mouse model. They dissociated germ cells using 0.1% collagenase and 0.25% trypsin in two steps. Okutsu et al. [3] performed xenogeneic spermatogonia transplant from rainbow trout to salmon (*Oncorhynchus masou*), obtaining germ cells by dissociating rainbow trout testes with 0.5% trypsin in PBS. The donor-derived germ cells colonized embryonic gonads and produced functional gametes after migrating into the peritoneal cavity. Psenicka et al. [16] used 0.3% trypsin in PBS at 23 °C for dissociation of Siberian sturgeon testicular and ovarian cells. Shikina et al. [19] demonstrated disruption of some membrane proteins by trypsin, causing a reversible decrease in mitotic activity and reduced adhesiveness. Gonad cells digested with trypsin presented higher growth rate than collagenase H, dispase II and trypsin-EDTA, but the yield of cells dissociated by other enzymes showed slight variation depending on species and gonad stage. Gonad protein composition likely undergoes changes during gonad development. Further studies are required to confirm the effect of trypsin on sturgeon germ cell migration ability after short term culture. Germ cells proliferated at 17 °C, 21 °C, and 25 °C, but 21 °C was more effective than 17 °C and 25 °C, in agreement with the optimal temperature for sturgeon cells reported by Grunow et al. [20].

Increasing concentration of FBS was associated with increased mitotic activity of somatic cells. Proliferation of germ cells decreased with increased culture duration, suggesting that gonad somatic cells depleted the available nutrients and growth factor.

Gonad somatic cells may be involved in the regulation of germ cell survival, proliferation, and differentiation under physiological and pathological conditions [21]. We attempted to assess the effects on germ cells of sturgeon gonad somatic cells as feeder cells. In mammals, feeder cells have been commonly used to culture cells at low densities [22]. Some studies have shown that, when co-cultured with Sertoli cells, germ cell maintenance is lower than with other feeder cell types in mice [23]. Rat spermatogonia show extended propagation with Sertoli cells compared to with Sandos inbred mice embryonic fibroblasts feeders [24]. Feeder cells have been used for induction of cell differentiation in fish [25]. In our study, sturgeon germ cells attached and proliferated during 14 d culture, but showed reduced mitotic activity after 14 d, shedding the feeder layer gradually, suggesting that gonad somatic cells may be not optimal for germ cell proliferation and amplification in sturgeon. Whether these feeder cells can induce differentiation needs further investigation. We also investigated the propagation of germ cells under feeder-free culture conditions and found higher proliferation maintained longer than for cells cultured with feeders. This possibility that purified germ cells tend to retain initial characteristics during in vitro culture suggests that there may be cross-talk between germ cell clones.

With or without feeder cells, germ cell propagation continued no longer than 21 d. In an in vitro culture system, supplemental growth factor is essential for long-term culture and self-renewal of germ cells [26,27,28]. Recent research in zebrafish has reported that recombinant mammalian growth factors containing bFGF and insulin-like growth factor 1 (IGF-1) promoted spermatogonia proliferation in an in vitro culture system. In Nile tilapia, Tokalov and Gutzeit [29] observed that IGF and hCG induced spermatogonia proliferation in vitro. In rainbow trout, studies showed that spermatogonia proliferation could be enhanced by IGF, follicle-stimulating hormone (Fsh), growth hormone, and FGF-2. To maintain germ cell propagation of for a longer period, we added a cocktail of soluble growth factors. The addition of LIF and GDNF induced the highest mitotic activity. Sturgeon serum also significantly increased germ cell proliferation, and was associated with proliferation extension for 21 d. Leukemia inhibitory factor is reported to be an essential growth factor that supports SSC self-renewal in mice [30]. In mammalian testis, Fsh treatment is reported to stimulate the expression of GDNF from Sertoli cells, which binds GDNF family receptor alpha-1 (Gfra1) in undifferentiated spermatogonia A, enhancing their self-renewal and maintenance [31,32,33]. In fish, expression of Gfra1 has been observed in Nile tilapia [34], dogfish [35], and rainbow trout [36] type A spermatogonia. However, GDNF pathways in teleosts differ from that in mammals. Shikina and Yoshizaki [7] demonstrated that neither rat GDNF nor rat GFRA1-Fc fusion protein enhanced the proliferation of rainbow trout spermatogonia in in vitro culture systems. Expression of *gdnf* has been observed in germ cells but not in Sertoli cells; hence, in rainbow trout, GDNF most likely acts as an autocrine factor [37]; whereas, in zebrafish spermatogonia in vitro culture, recombinant human GDNF was observed to promote spermatogonia proliferation. In dogfish, GDNF could promote self-renewal of potential spermatogonia stem cells in culture [38]. It would be interesting to determine how these factors influence the in vitro proliferation of sturgeon germ cells. In previous studies, serum of fish including carp and rainbow trout was reported to promote self-renewal of zebrafish or medaka stem cells [5,39]. However, an homologous fish serum for germ stem cell culture has not been reported. Use of serum from the same species might prevent rejection. Sturgeon serum may be a potential alternative to LIF and GDNF.

Barnes and Enat [40,41] reported that FBS contains complex unidentified materials such as plasma proteins, polypeptides, growth factors, hormones, and binding proteins that preserve cell survival when they attach to flasks [40,41]. It may also contain inhibitors of certain tissue-specific cells. An in vitro study of goat germ cell culture showed that high serum concentration may have detrimental effects on germ cell expansion and simultaneously stimulate extensive germ cell growth. Spermatogonia stem cells proliferated for seven days when supplemented with 1% FBS, but higher FBS levels impaired spermatogonia propagation [42]. Spermatogonia stem cell differentiation can be prompted by unidentified components in serum [40]. In our experiment, in the serum-free medium, mitotic activity of germ cells remained elevated for at least 34 d in vitro, and the growth of somatic cells was significantly inhibited.

qPCR analysis revealed no significant differences in marker gene expression before and after 10 d culture, indicating that the sturgeon germ cell population was able to expand in vitro while maintaining original characteristics. We also assessed whether cultured germ cells developed in recipients. Since no effective RNA primers distinguishing sterlet from Russian sturgeon are available, we injected pure cultured germ cells (91%) from sterlet into Russian sturgeon larvae and assessed with a sterlet-specific DNA primer. Results showed that cultured cells maintained their activity to incorporate into recipient gonads. Although it is uncertain whether the cells were germ cells or somatic cells, 91% of the transplanted cells were germ cells, and we suggest it is more likely that the cells developed in recipients were germ cells. Whether normal offspring can be obtained from cultured germ cells in our culture system requires further studies.

In conclusion, we established effective culture conditions that improved the mitotic activity of germ cells and maintained their survival for at least 40 d in culture with stable proliferation in vitro. The results offer useful details for culturing germ cells in sturgeon, representing a first step towards the establishment of germ cell lines that will be useful in germ cell xenotransplantation.

## 5. Conclusions

Enriched germ cells cultured in a serum-free medium proliferated 40 days in vitro. Cultured cells showed development in recipient genital ridges. The culture conditions that we describe could supply germ cells for transplantation and surrogate production of sturgeon and for other in vitro studies.

## Figures and Tables

**Figure 1 animals-09-00106-f001:**
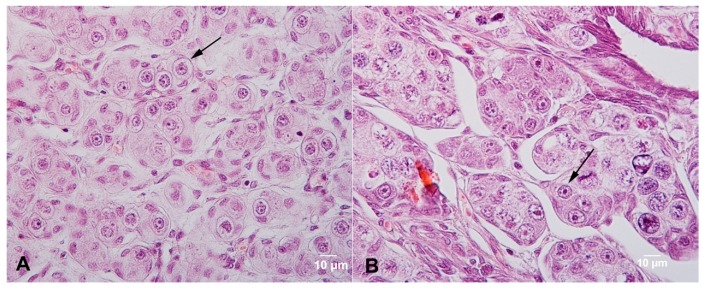
Sterlet *Acipenser ruthenus* gonads. (**A**) Sterlet testis at maturity stage II (10-month-old); (**A**) showing spermatogonia (black arrow) and Sertoli cells; (**B**) Sterlet ovary in maturity stage II (10-month-old); (**B**) showing oogonia (black arrow) and early oocytes.

**Figure 2 animals-09-00106-f002:**
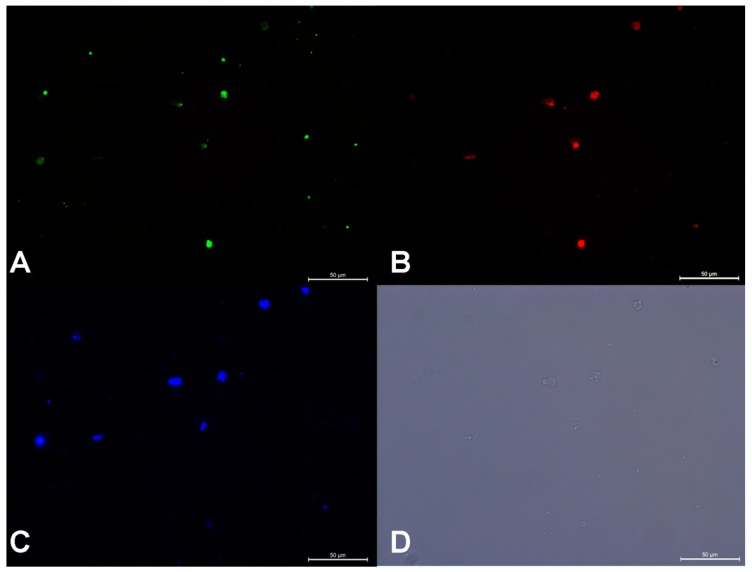
Immunofluorescence showing co-localization of neonatal cells at 10-days culture. Gonads were digested with trypsin and cultured at 21 °C in L-15 medium and 10% fetal bovine serum (FBS) (*v/v*). Cells were stained green for BrdU protein (**A**), red for Vasa protein (**B**) and blue for 4′,6-diamidino-2-phenylindole (DAPI) (**C**). (**D**) White light.

**Figure 3 animals-09-00106-f003:**
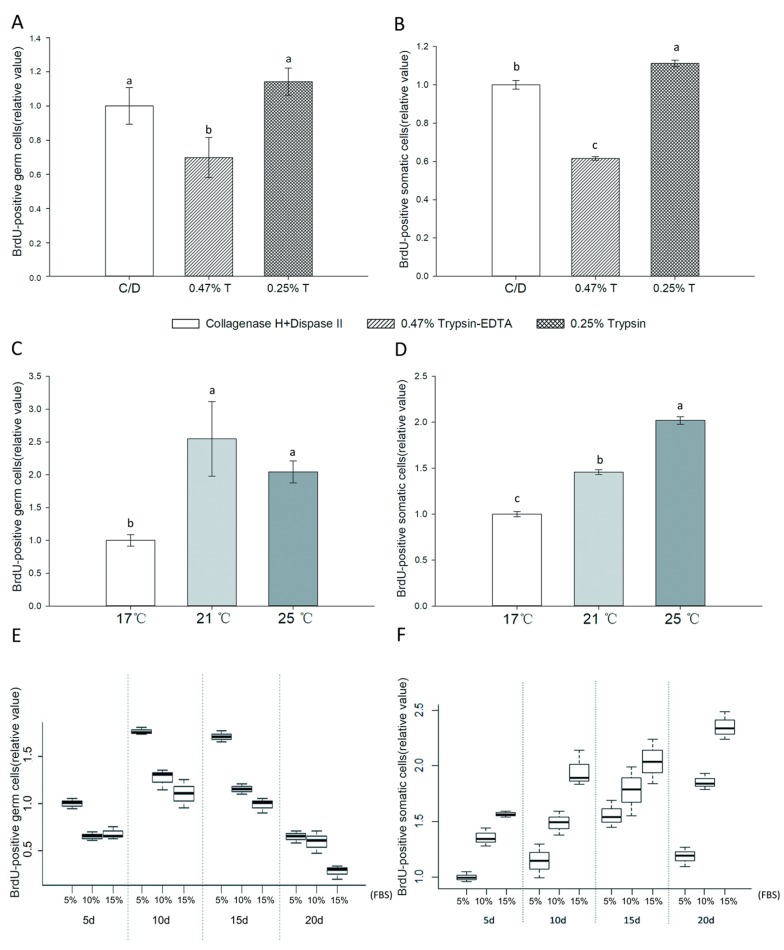
Influence of enzymatic dissociation (**A**,**B**), culture temperature (**C**,**D**), and FBS (5–15%) (**E**,**F**) on BrdU incorporation into Dabry’s sturgeon germ cells. (**A**,**B**): Gonads were dissociated with 2 mg/mL collagenase H and 500 U/mL Dispase II (C/D); 0.25% trypsin (0.25% T), or 0.47% trypsin– Ethylenediaminetetraacetic acid (EDTA, 0.47% T-EDTA). Cells were cultured in L-15 medium (pH 8.0) supplemented with 15% FBS for 10 d at 25 °C. (**C**,**D**) Gonad was dissociated with 0.25% trypsin and cultured in L-15 medium supplemented with 15% FBS (pH 8.0) for 10 d at 17, 21, and 25 °C. (**E**,**F**) Gonad was dissociated with 0.25% trypsin and cultured in L-15 medium (pH 8.0) under selected concentrations of FBS (5–15%) at 21 °C. BrdU assays were performed every 5 d. Data are shown as mean ± SEM (*n* = 4) of relative values for germ cells and somatic cells by immunofluorescent co-localization of BrdU and Vasa. The values of collagenase H, Dispase II (**A**,**B**), 17 °C (**C**,**D**), and 5% FBS (5 d) (**E**,**F**) were defined as the relative value 1.0. Values with different lowercase letters are significantly different (*p* < 0.05).

**Figure 4 animals-09-00106-f004:**
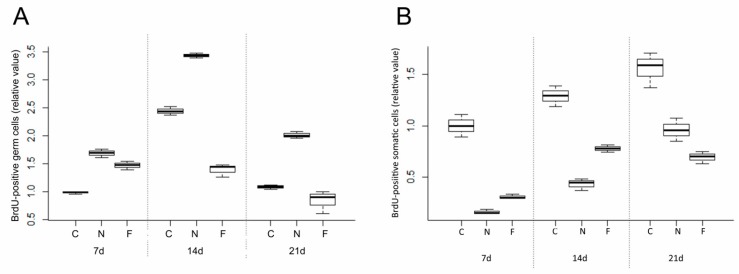
Effects of somatic cells on mitotic activity of Dabry’s sturgeon germ cells (**A**) and somatic cells (**B**). The figure shows mean ± SEM (*n* = 3) of total mitotic activity under control (C), feeder-free (N), and feeder layer conditions (F). The BrdU assay was performed on days 7, 14 and 21 of culture. Data are shown as the mean ± SEM (*n* = 3). The values for 7-d control were defined as the value relative to 1.0. Values with different lowercase letters were significantly different according to two-way ANOVA (*p* < 0.05).

**Figure 5 animals-09-00106-f005:**
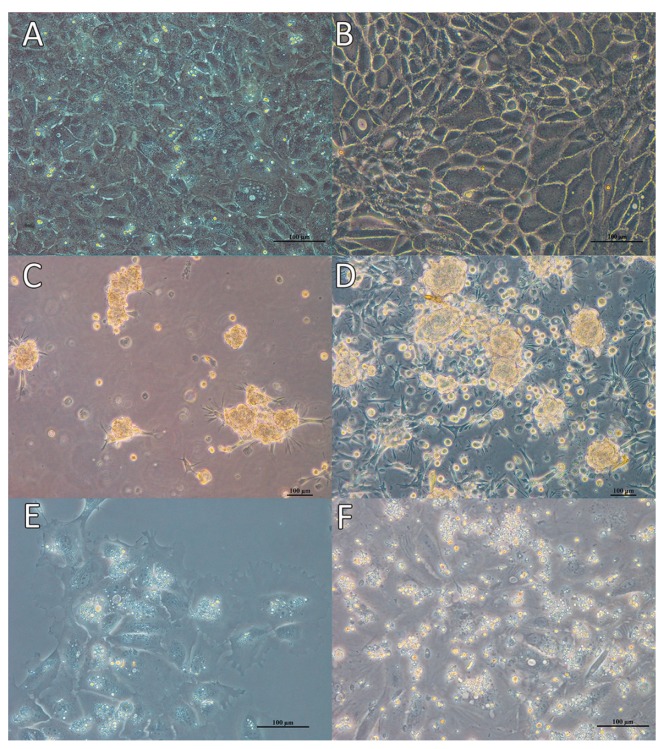
Morphology of Dabry’s sturgeon gonad cells (**A**,**B**) and purified germ cells cultured without (**C**, **D**) or with feeder cells (**E**,**F**), after 3 d (**A**,**C**,**E**) and 14 d (**B**,**D**,**F**).

**Figure 6 animals-09-00106-f006:**
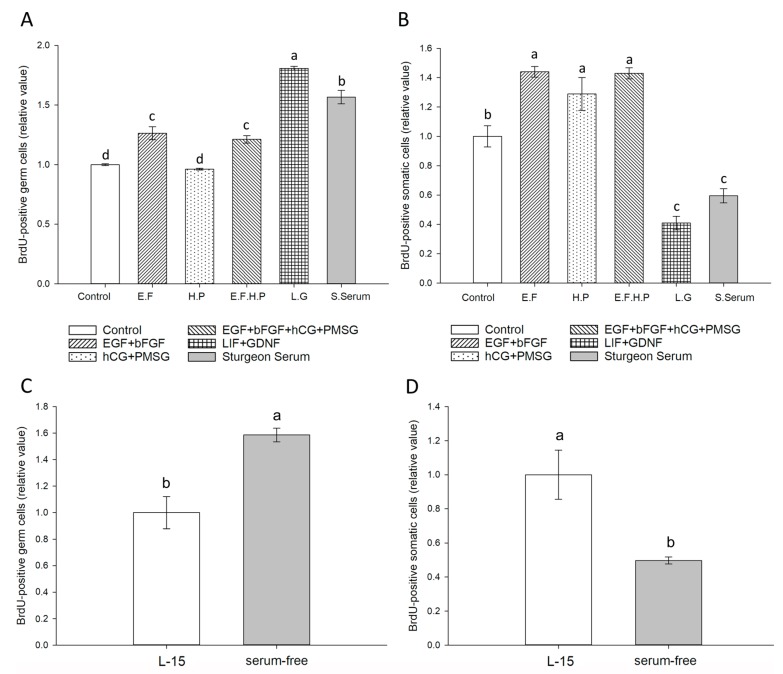
Effects of selected soluble growth factors on mitotic activity of Dabry’s sturgeon germ cells (**A**) and somatic cells (**B**). The number of germ cells was counted on Day 10. The somatic cell mitotic activity after addition of EGF and bFGF (E.F), hCG and PMSG (H.P), EGF/bFGF/hCG and PMSG (E.F.H.P.), LIF and GDNF (L.G.), and sturgeon serum (S. Serum) is shown. Data are shown as the mean ± SEM (*n* = 3). The control values (**A**,**B**) were defined as the relative value 1.0. Values with different lowercase letters are significantly different (*p* < 0.05) according to one-way ANOVA. Effect of replacement of FBS on mitotic activity of germ cells (**C**) and somatic cells (**D**). The number of germ cells were counted on Day 10. Cells were cultured at 21 °C in L-15 medium (pH 8.0) supplemented with 25 mM HEPES, antibiotics, 5% FBS, and 25 ng/mL LIF and GDNF (L-15, white bar) and StemPro-34 Serum-Free Medium with supplement (serum-free, gray bar). Data are shown as mean ± SEM (*n* = 3). The values of L-15 medium were defined as the relative value 1.0. Values with different lowercase letters are significantly different (*p* < 0.05) according to *t*-test.

**Figure 7 animals-09-00106-f007:**
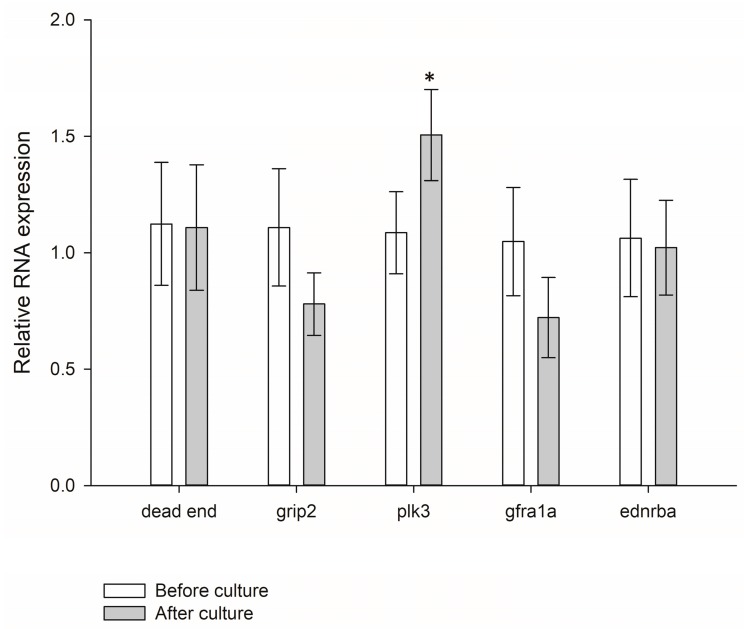
Expression of *dead end*, *grip2 plk3, gfra1a*, and *ednrba* in germ cells before and after 10-d culture. Data are shown as mean ± SEM (*n* = 3). Differences were considered significant at *p* < 0.05. The asterisk indicates significant difference of gene expression compared to before culture according to paired *t*-test.

**Figure 8 animals-09-00106-f008:**
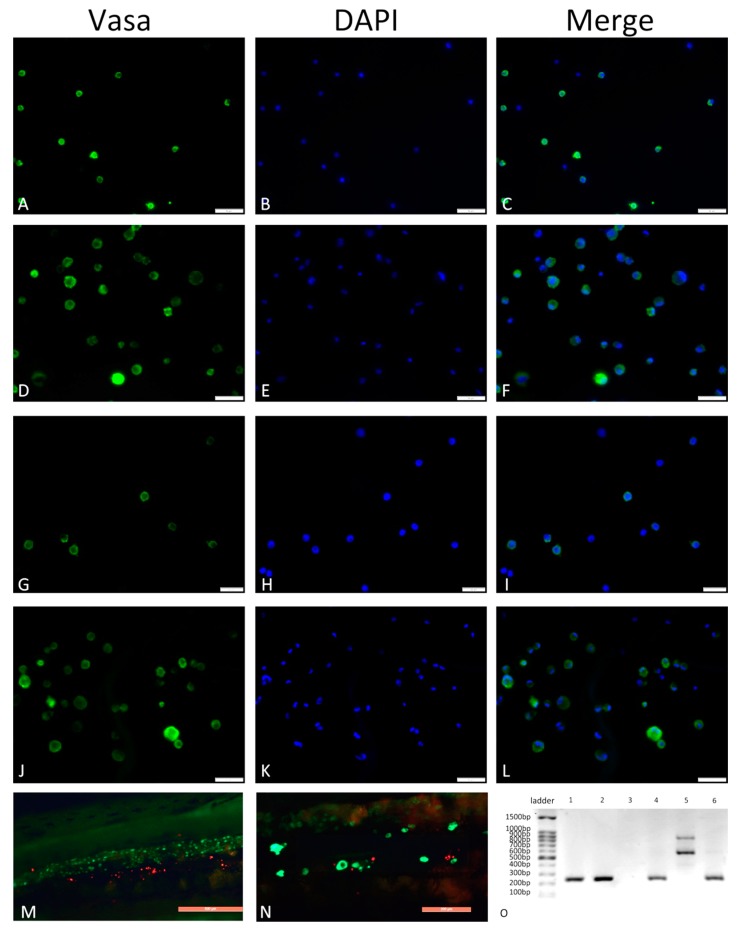
Immunofluorescence of sterlet germ cells in serum-free medium. Purified spermatogonia before culture (**A**–**C**) and before transplant after 40 d culture (**D**–**F**). Purified oogonia before culture (**G**–**I**) and before transplantation after 40d culture (**J**–**L**). Russian sturgeon larvae transplanted with cultured sterlet germ cells 40 dpt (**M**,**N**). Scale bars: 50 μm (**A**–**F**,**J**–**L**), 20 μm (**G**–**I**), 500 μm (**M**), 200 μm (**N**). Identification of sterlet from recipients by presence/absence of band (**O**). 1–3: Amplification of species-specific primer 247_ARp (sterlet positive). 1 = recipient gonads; 2 = normal sterlet fin; 3 = normal fin. 4–6: Amplification of species-specific primer 247_ARn (sterlet negative Russian sturgeon). 4: recipient’s gonads. 5: normal fin from sterlet. 6: normal fin from Russian sturgeon.

**Table 1 animals-09-00106-t001:** Quantitative polymerase chain reaction (PCR) primer sequences for germ-line specific genes (*dead end*, *gfra1a*, *grip2*, *plk3* and *ednrba*).

Name	Sequence (5′-3′)	Orientation	Length bp
Ardnd1	AAACGTGAGGCACGGGTATT CCTGGATCGGTATCCACAGC	Forward Reverse	705
cds(a)_gfra1a	GGAAGTGGGAACAGGGAAGA GGGTTTGGGTGCTAGATTTGT	Forward Reverse	123
grip2	TGCTGAAGAATGTGGGCGA CCCTCTCAACACGAAGCCA	Forward Reverse	149
plk3	ACCCGAGTCAGATGTGTGGT AGCAGGAAGGGAGAGGAAGT	Forward Reverse	148
ednrba-201_CDS	TTAGGCGCTTCCGAGACTAC CCGGGTTCATGGTTTTGGG	Forward Reverse	81

**Table 2 animals-09-00106-t002:** Primers used for the DNA amplification of sterlet and Russian sturgeon.

NAME	SEQUENCE (5′-3′)
247_ARp	TAAGGGTCCATGCATGCAG
247_ARn	TAAGGGTCCATGCATGCCT
247_uni	TTTTAGCTGCACCGTGGC

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
