# Peer review of "Optimization of In Vitro Culture Conditions of Sturgeon Germ Cells for Purpose of Surrogate Production"

_animals, 2019, doi:10.3390/ani9030106_

Round 1
Reviewer 1 Report
NA
Reviewer 2 Report
Dear Authors and editors
Appreciate the feedback results.
I would consider this paper ready for publishing.
This manuscript is a resubmission of an earlier submission. The following is a list of the peer review reports and author responses from that submission.
Round 1
Reviewer 1 Report
Review of paper.
Optimization of in vitro culture conditions of sturgeon germ cells for
purpose of surrogate production
Authors: Xuan Xie, Ping Li, Martin Pšenička, Huan Ye, Christoph Steinbach,
Chuangju Li, Qiwei Wei *
This paper explores in vitro culture conditions for sturgeon germ cells and finally explores the possibility for xenotransplantation.
Major points
This paper explores possibilities sturgeon conservation and is therefore an important contribution to this field. The paper manages to test many conditions for cell culture of germ cells.
However, there are some major points that needs to be addressed:
(1) It would have been nice to see some more proof of their BrdU counts such as histological illustrations of how the BrdU counts were performed, marking both somatic and germ cells in the histological sections.
(2) Another major point is the analysis of the xenotransplantation. How do the authors know that it is the germ cells and not the somatic cells that have survived in the host?
(3) Also, there is a nice discussion on different conditions for germ cell cultivation.
However, no discussion is found on the xenotransplantation.
(4) I suggest removing the xenotransplantation part from the manuscript or alternatively add more data on xenotransplantation and discuss these.
(5) The paper suffers from numerous grammatical errors and I would strongly suggest English improvement in particular of abstract and introduction.
Author Response
Response to Reviewer 1 Comments
Point 1: It would have been nice to see some more proof of their BrdU counts such as histological illustrations of how the BrdU counts were performed, marking both somatic and germ cells in the histological sections.
Response 1: Thanks you so much for this helpful comments. As you suggested, a figure is added in text, showing the morphology of cells which incorporated BrdU and performed vasa-positive signals.
Point 2:Another major point is the analysis of the xenotransplantation. How do the authors know that it is the germ cells and not the somatic cells that have survived in the host?
Response 2:Thank you so much for this valuable comments. So far any effective RNA primers distinguishing sterlet and Russian sturgeon have not been found, but we have very sensitive DNA primer to detect sterlet and Russian sturgeon. We injected pure cultured germ cells (91%) from sterlet into Russian sturgeon larvae and checked with sterlet DNA specific primer. Results showed there were some cultured cells maintained their activity to incorporate into recipient gonads. Although it’s uncertain whether the cells were germ cells or somatic cells, we used the most transplanted cells (91%) were germ cells and therefore we may predict that it’s more likely that the developed cells in recipients were germ cells.
Point 3:Also, there is a nice discussion on different conditions for germ cell cultivation.
However, no discussion is found on the xenotransplantation.
Response 3:Thank you for the suggestion. We added the discussion about xenotransplantation from line 456 to 465.
Point 4:I suggest removing the xenotransplantation part from the manuscript or alternatively add more data on xenotransplantation and discuss these.
Response 4:Thank you so much for this valuable comments. We showed the ratio of germ cells before transplantation and discussed those in the manuscript.
Point 5:The paper suffers from numerous grammatical errors and I would strongly suggest English improvement in particular of abstract and introduction.
Response 5:Thank you very much to point out the grammatical issues in our manuscript. According to your comments, we polished the manuscript with a professional assistance in writing, conscientiously.

Reviewer 2 Report
Concerns from this reviewer:
Since Brdu incorporation and Vasa immunocytochemistry were used as a main approach to study germ cell culture, the authors, in addition to the graphic results, should show some fluorescent photos of these results to help readers understand this approach.
Font size in Figure 2, 3, 5 are too small to read and need to be improved.
Scale bars were incorrectly labeled or described such as Fig 1 (10 or 50 μm?) and 4 (μm or mm?). Please check all of them.
Author Response
Response to Reviewer 2 Comments
Point 1: Since Brdu incorporation and Vasa immunocytochemistry were used as a main approach to study germ cell culture, the authors, in addition to the graphic results, should show some fluorescent photos of these results to help readers understand this approach.
Response 1: Thank you so much for this valuable advice. As you suggested, a figure is added in text, showing the morphology of cells which incorporated BrdU and performed vasa-positive signals.
Point 2:Font size in Figure 2, 3, 5 are too small to read and need to be improved.
Response 2:Thank you for this comments. We have modified font size in the new manuscript.
Point 3:Scale bars were incorrectly labeled or described such as Fig 1 (10 or 50 μm?) and 4 (μm or mm?). Please check all of them.
Response 3: Thank you so much for your careful reading of our manuscript. We are very sorry for our fault statment in the legends. We have corrected it in the new manuscript.